# Alexithymia in Alzheimer’s Disease

**DOI:** 10.3390/jcm10010044

**Published:** 2020-12-25

**Authors:** Eva Mª Arroyo-Anlló, Corinne Souchaud, Pierre Ingrand, Jorge Chamorro Sánchez, Alejandra Melero Ventola, Roger Gil

**Affiliations:** 1Department of Psychobiology, Neuroscience Institute of Castilla-León, University of Salamanca, 37007 Salamanca, Spain; 2Department of Neurology and Neuropsychology, University Hospital, CHU La Milétrie, 86000 Poitiers, France; C.Souchaud@chu-poitiers.fr; 3Department of Biostatistics, University of Poitiers, 86000 Poitiers, France; pierre.ingrand@univ-poitiers.fr; 4Faculty of Psychology, Pontifical University of Salamanca, 37002 Salamanca, Spain; jchamorrosa@upsa.es (J.C.S.); amelero@cop.es (A.M.V.); 5Department of Neurology, University Hospital, 86000 Poitiers, France; roger.gil@wanadoo.fr

**Keywords:** emotion, neurodegenerative diseases, Alzheimer, dementia, language, depression, awareness

## Abstract

Alexithymia is widely recognized as the inability to identify and express emotions. It is a construct which consists of four cognitive traits such as difficulty in identifying feelings, describing feelings to others, externally oriented thinking, and limited imaginative capacity. Several studies have linked alexithymia to cognitive functioning, observing greater alexithymia scores associated with poorer cognitive abilities. Despite Alzheimer’s disease (AD) being a neurodegenerative pathology characterized by cognitive troubles from the early stages, associated to behavioral and emotional disturbances, very few investigations have studied the alexithymia in AD. These studies have shown that alexithymia scores—assessed with Toronto Alexithymia Scale (TAS)—were greater in AD patients than healthy participants. The objective of the study was to investigate if the alexithymia was present in patients with mild AD. We hypothesized that the AD group would show more alexithymia features than the control group. We evaluated 54 subjects, including 27 patients diagnosed with mild AD and 27 normal healthy controls, using the Shalling Sifneos Psychosomatic Scale (SSPS-R) and a neuropsychological test battery. Using non-parametric statistical analyses—Wilcoxon and Mann–Whitney *U* tests—we observed that the SSPS-R scores were similar in the AD and control groups. All participants showed SSPS-R scores below to 10 points, which means no-alexithymia. We did not find significant correlations between SSPS-R scores and cognitive variables in both groups (*p* > 0.22), but we observed a negative association between name abilities and alexithymia, but it does not reach to significance (*p* = 0.07). However, a significant correlation between SSPS-R score and mood state, assessed using Zerssen Rating Scale, was found in both groups (*p* = 0.01). Because we did not find a significant difference in the alexithymia assessment between both subject groups, pot hoc analyses were computed for each item of the SSPS-R. We made comparisons of alexithymic responses percentages in each SSPS-R item between AD and control groups, using Fisher’s test. We observed that AD patients produced more alexithymic responses in some items of SSPS-R test than the control group, particularly about difficulties to find the words to describe feelings, as well as difficulties of imagination capacity and externally oriented thinking. The present results do not confirm our hypothesis and they do not support the results of previous studies revealing great alexithymia in AD.

## 1. Introduction

Alexithymia means “no words for feeling” from the Greek. At the end of the 1940s, the concept was referenced by Ruesch [1] and in 1972, Peter Sifneos [2] introduced this term to describe the inability to find appropriate words to describe one’s feelings in patients with psychosomatic disorders. In 1977, Freyberger [3] suggested alexithymia as a personality trait and a state-like secondary phenomenon after somatic disease or psychological stress [3,4,5,6,7]. 

In a strict sense, alexithymia is widely recognized as the inability to identify and express emotions, as well as difficulty in communicating them [2,8,9]. Currently, it is described as a difficulty in emotional processing rather than a defensive process, but the comprehensive models of this construct still remain as having no consensus ([10], for review). It is well recognized as a dimensional cognitive-affective style that influences emotional regulation with impacts on health. Alexithymia is a construct which is comprised of four components such as: difficulty identifying feelings, describing feelings to others, externally oriented thinking, and limited imaginative capacity or restricted fantasizing. 

Actually, there are three main scales to assess alexithymia: the Beth Israel Questionnaire (BIQ) [2], the Shalling Sifneos Psychosomatic Scale (SSPS) [11], and the Toronto Alexithymia Scale (TAS) [12], but other measures have been used such as M.M.P.I. Alexithymia Scale [13], projective tests... In 1973, Sifneos introduced the first questionnaire, Beth Israel Questionnaire (BIQ), for screening of alexithymic characteristics [2], The BIQ is a 17-items forced-choice questionnaire completed by the interviewer. Later, Sifneos [11] created the Shalling Sifneos Psychosomatic Scale (SSPS) which is formed by 20 questions with dichotomous responses (yes/no). It lets us analyse 3 factors: the difficulty of expressing feelings, the importance of feelings, as well as the daydreaming and introspection. Alexithymia can also be assessed using the Toronto Alexithymia Scale (TAS) [12], which is a 26-items self-report measure rated on a five-point Likert scale. It comprises three subscales assessing three facets of alexithymia: difficulty in identifying feelings, difficulty in describing feelings and an externally oriented analytic mode of thinking.

Although initially described in the context of psychosomatic illness, alexithymic characteristics may be observed in patients with a wide range of psychiatric disorders. However, alexithymia also seems to be a common feature of neurological disease, with most evidence available for patients with traumatic brain injury, stroke, and epilepsy, associated frequently with mood disorders such as depression and anxiety (see review [14,15]). 

Concerning neurodegenerative diseases, a small number of studies have investigated the prevalence of alexithymia in patients with multiple sclerosis (MS) [16,17,18,19]. They observed a prevalence of 13–30% in the patient groups, depending on the study. However, mostly studies reported that mood state deficits (anxiety and/or depression) were associated with alexithymia [16,17,19]. Alexithymia has also been reported frequently in patients suffering from Parkinson’s disease (PD) [20,21,22], but it is still not clear whether it is linked to depression [20,23,24,25,26,27]. In a recent work, Assogna et al. [15] studied the alexithymia in two variants of Progressive Supranuclear Palsy (PSP), Richardson’s syndrome (PSP-RS), and PSP with predominant parkinsonism (PSP-P) compared to PD. Alexithymic symptoms differentiate PSP-RS group from PD group but not between the two subtypes of PSP. They also observed that alexithymia was identifiable very early in PSP patients. Further, alexithymia in PSP seemed to be predicted by the presence of depression. 

Furthermore, several studies have linked alexithymia to cognitive functioning ([28,29,30,31], for a review). They have observed greater alexithymia scores associated with poorer cognitive abilities, particularly it being associated with language and executive troubles [32,33,34,35,36]. Other studies have also found that higher alexithymia correlated with poorer attention and memory [37,38]. 

Despite Alzheimer’s disease (AD) being a neurodegenerative pathology characterized by cognitive troubles from the early stages, associated with behavioral and emotional disturbances [39,40,41,42,43], very few investigations have studied alexithymia in AD. Thus, little is known about the expression and verbalization of emotions from patients with AD. Recently, three research projects have specifically focused on alexithymia in AD, to our knowledge [44,45,46]. All studies have shown that alexithymia scores, assessed with TAS, were elevated in AD patients. Besides, Sturm and Levenson [44] found alexithymia scores were positively correlated with behavioral deficits (apathy and informant distress) in patients with dementia and negatively correlated with the grey matter volume of the right pregenual anterior cingulate cortex, a region of the brain that is thought to play an important role in self and emotion processing. The study of Yuruyen and his team [45] assessed alexithymia with TAS in patients and they observed significantly greater alexithymia in AD and mild cognitive impairment (MCI) groups than that of individuals with memory loss complaints in each TAS factor, but not significant difference between AD and MCI patients. The preliminary results of Smirni et al. [46] reported greater alexithymia total scores in AD and MCI patients than in healthy subjects, but only significant higher scores on the TAS factor difficulty in identifying feelings. 

On the basis of this background, the aim of the study was to investigate quantitatively and qualitatively, if the alexithymia was present in AD patients. Here, we hypothesized that patients with AD show more alexithymia features than healthy control population. 

## 2. Materials and Methods

### 2.1. Subjects

We studied 54 subjects, including 27 patients diagnosed with AD and 27 normal healthy controls. Twenty-seven individuals with a clinical diagnosis of probable AD were recruited at the Department of Neurology and Neuropsychology, University Hospital, CHU La Milétrie at Poitiers, France. All lived at home with a family caregiver. They had a history of progressive decline in intellectual function without focal motor or sensory features. To exclude other possible causes of dementia, appropriate laboratory tests were performed, and these gave normal results. No findings incompatible with a diagnosis of AD were found in the electroencephalogram, electrocardiogram, or chest X-ray in any of the patients. Brain CT scan revealed mild cortical and central atrophy, but no other pathology. All Alzheimer patients met the diagnostic criteria of the “NINCDS—ADRDA Work Group” and of the DSM-5. All had a score of less than 5 on the Hachinski Ischemic Scale. According to the Clinical Dementia Rating System (CDR) [47] and Mini-Mental State Examination (MMSE) [48], all patients had mild AD. All AD subjects were taking anti-dementia medication during the study, in particular anti-acetyl cholinesterase treatments. Only the AD patients whose verbal comprehension (assessed by the MMSE three-stage command) were equal to or above 2 were included in the study. 

Controls primarily comprised of volunteers or were members or visitors at the University Hospital in Poitiers (France) whose results in a neurological examination were normal, had a CDR score [47] equal to zero and scored 28/30 or higher on the MMSE [48].

### 2.2. Neuropsychological Assessment

A comprehensive battery of neuropsychological tests was administered to all participants to evaluate the cognitive capacities, mood state and alexithymia.

On one hand, cognitive functions were evaluated using Mini-Mental State Evaluation (MMSE) [48] to assess the general mental state. Episodic memory was explored using the verbal memory test designed by Grober and Buschke [49], consisting of a word list learning and free/cued recall. We also used the DO80 verbal naming test [50], which assesses pictures naming to evaluate language abilities. In addition, we assessed the mood state, using the Zerssen Rating Scale [51].

On the other hand, we completed the neuropsychological assessment using the Shalling Sifneos Psychosomatic Scale-Revised (SSPS-R) [52] to evaluate the existence of alexithymia features in the participants. We chose SSPS-R because it is adapted to dementia populations, as well as it is completed and helped by the interviewer. Besides, it consists in 20 specific sentences about daily behaviours and feelings of the subjects. The subjects with a SSPS-R score between 10 and 20 points were considered as alexithymic.

### 2.3. Procedure

All participants were assessed individually in only one session with the same order of testing: first, it was for the evaluation of cognition and after, exclusively for testing mood state and alexithymia. The session took between 60–90 min to complete.

### 2.4. Ethics

Written, informed consent was obtained from all the included patients. The study was approved by The Regional Committee for Research Ethics, CHU Poitiers.

### 2.5. Data Analysis

Data was analysed using the Statistical Package SAS—software (version 9.2, SAS Institute Inc, Cary, NC, USA). In this study an α level of 0.05 was selected for statistical significance. To check whether there were significant differences between the two groups as a function of age and education years, we used Wilcoxon two sample test and in the case of gender, we used X^2^ test of Fisher.

To compare the mean scores of cognitive and alexithymia abilities between participant groups, non-parametric Mann–Whitney *U* and Wilcoxon tests were used. The fisher test was also used to compare the alexithymic responses proportion in each SSPS-R item between both groups. Within-group (AD and healthy control groups) correlations were also planned for SSPS-R total scores and clinical/cognitive variables, using the Spearman coefficient.

## 3. Results

### 3.1. Demographic and Neuropsychological Characteristics

Fifty-four subjects participated in the study: an experimental group with twenty and seven AD patients, and a control group with twenty and seven healthy participants. Concerning the experimental AD group (19 women and 8 men), it had a mean age (±SD) of 79.17 years old (±7.43), a mean education years (±SD) of 8.94 (±1.64), and a MMSE score (±SD) of 20.28 (±4.61). The AD group showed a mild disease, assessed with MMSE and CDR (1.4 ± 0.23) and the mean (±SD) duration of the disease was 4.1 (±1.22) years. Concerning the control group (13 women and 14 men), the mean age (±SD) was 76.59 years old (±9.43). It had mean education years (±SD) of 9.26 (±1.11), and a MMSE score (±SD) of 29.18 (±1.13). There were no significant differences in gender (*p* = 0.062), age (*p* = 0.98), and education years (*p* = 0.84). The means and standard deviations of the different demographic characteristics of the AD and the control groups are shown in Table 1.

In relation to the battery of neuropsychological tests administered to both groups, the means and standard deviations of the neuropsychological tests are shown in Table 2. We found significant differences between two participant groups for the general mental state evaluated by MMSE test [48] (*p* = 0.0001) and for language capacity assessed by the DO80 verbal naming test [50] (*p* = 0.032). Concerning the memory capacities evaluated by Grober and Buschke verbal memory [49], analysis also revealed significant differences between both groups (*p* < 0.01).

In regard to mood state assessed by the Zerssen Rating Scale [51], there were no significant differences between the groups; AD and control groups did not have depressive disorders (mean ± SD: 11.63 ± 7.39 and 12.18 ± 6.12, respectively; *p* = 0.066).

### 3.2. Alexithymia Assessment

The alexithymia was assessed using the Shalling Sifneos Psychosomatic Scale (SSPS-R) [52] and the results are reported in Table 2. All participants showed SSPS-R scores below to 10 points. The SSPS-R scores had similar means (±SD) in the AD and control groups (5.44 ± 2.75 and 5.35 ± 2.76, respectively; *p* = 0.93). Besides, analyses were computed to examine correlations between SSPS-R scores and cognitive and mood variables within each participant group. We did not find significant correlations between SSPS-R scores and cognitive variables in both groups (*p* > 0.22), but we observed a negative association between name abilities and alexithymia, but it does not reach significance (*p* = 0.07). Besides, a significant correlation between SSPS-R score and mood state, assessed using Zerssen Rating Scale [51], was found in both groups (*p* = 0.01). Table 3 shows the results of these correlations.

Because we did not find a significant difference in the alexithymia assessment between both subject groups, pot hoc analyses were computed for each item of the SSPS-R (Table 4 and Figure 1). We made comparisons of alexithymic responses percentages in each SSPS-R item between AD and control groups, using Fisher’s test. We observed that the most frequent alexithymic responses made by participants of AD group decreasingly were in the following item numbers of SSPS-R: 6 (“I spend much time daydreaming “), 1 (“It is easy to describe symptoms or complaints rather than feelings”), 5 (“I lack imagination”), 19 (“I dream rarely “) and 3 (“It is hard to use words to describe feelings”) and in the items numbers 6, 10 (“When in trouble I don’t like to act”), 1, 7 (“When I am mad I don’t think, I take action”), and 11 (“I like to be precise and to describe everything in detail “) by participants of the control group. The least frequent alexithymic responses made by individuals of AD group increasingly were in the following item numbers of SSPS-R: 2 (“It is important to find out how one feels about people”) and 4 (“I think that feelings are what make life worthwhile”) and in the 2, 4, 12 (“I don’t care to describe details but rather I prefer to examine how I feel”), 18 (“I cannot visualize circumstances which upset me”), and 20 (“I like people better than things”) items by individuals of control group. We found a significant difference of alexithymia responses in the following item numbers of SSPS-R: 3, 5, 7, 10, 11, 14, 15, 18, and 19 between both groups (*p* < 0.024).

## 4. Discussion

The general aim of this study was to determine whether an AD group could show alexithymia, using the SSPS-R test [52]. We hypothesized that patients with mild AD would show more alexithymia features than healthy control population, but it was not confirmed. All participants showed no alexithymia scores, considering all SSPS-R total scores were below 10 points. We suggest that the earlier stages of AD could not be associated with alexithymia. All participants showed very interested in emotions and feelings (items 2 and 4), but the most of AD and healthy participants answered that they had difficulties describing feelings (item 1) and they spent a short time daydreaming (item 6), which are the two most important characteristics of alexithymia in any comprehensive model of alexithymia [10]. A possible explanation about why the AD group did not show alexithymia could be that AD is associated with reduced insight into personal impairments from the early stage of disease [53,54,55,56], suggesting that the mental skills required to access personal emotional knowledge were in part supported by cognitive functions that process non-emotional material. However, the AD patients in our study explicitly communicated their difficulties to “find the words describing their emotions”. Regarding that alexithymia is also considered as a form of reduced insight into personal emotions, a limitation of our study could be not to have included an assessment of awareness, taking into account its great prevalence in people with AD in early phases of neuropathology [57,58,59]. Thus, it could have limited the potential inferences between reduction of self-awareness, cognitive decline, and alexithymia in AD. It would also be interesting to consider the self-consciousness factors (anosognosia, introspection, consciousness of body representation, of mood state...) in future research about alexithymia and neurodegenerative diseases, in order to improve the comprehension about alexithymia.

In addition, the literature about cognition and alexithymia revealed that alexithymia is linked to cognitive troubles [28,29,30,31,32,33,34,35,36,37,38,46]. However, we did not find any association between SSPS-R scores and cognitive capacities, such as those of memory or language, as well as general mental state. Nevertheless, alexithymia could be dependent on lexical stock assessment with the DO80 test, though the correlation did not reach significant effect. However, Sturn and Levenson [44] observed that alexithymia found in patients with dementia could not be accounted for by a general decline in language functioning. It would be interesting to have a bigger size of participant group, in order to verify if the language troubles could have an impact on alexithymic features in AD. Moreover, it has also been suggested that an interhemispheric functional disconnection is relevant to alexithymia, so that emotional information from the right hemisphere is not properly transferred to the language regions in the left hemisphere [14,57,60] which are damaged in early stages of AD. In addition, several studies found the non-verbal language aspects which are more dependent of right hemisphere functioning [61], were relatively preserved in mild AD [62,63].

However, when we compared each item of SSPS-R between AD and control groups, we found some significant different proportions of participants—among the subject groups—who expressed alexithymic responses in each item. Considering the alexithymic features in terms of Sifneos [2,11] more AD patients significantly expressed difficulties of describing feelings (item 3), and limited imaginative processes (items 5, 18, and 19) than the control group. We also found a greater proportion inside the control group which showed preference to resolve the troubles by action (items 7 and 10) than among the AD group. Furthermore, we observed preference to be oriented toward the outside in an important percentage in both groups (items 11 and 15), but more among the control group. Around half of the AD group avoid interacting with people (item 14) in a higher proportion than that of the control group.

The alexithymic cognitive style of Sifneos [2,11] is characterized by the following alexithymic features: (1) difficulty in identifying and describing feelings, (2) difficulty in distinguishing between feelings and bodily sensations related to emotional activation, (3) restrained and limited imaginative processes, adopting the guise of an impoverished fantasy, (4) a cognitive style oriented toward the outside, (5) poor interpersonal relationships, and (6) troubles resolution by action [64]. However, in terms of Sifneos [2,11] and Freyberger [3], we cannot suggest that AD group showed an alexithymic cognitive style, but we observed some greater alexithymia features in AD patients than in control participants, particularly concerning the difficulties in describing feelings and imagination abilities, which could be secondary to cognitive decline.

Moreover, we observed that the AD group produced responses which expressed a more passive attitude and internally oriented thinking than in the control group, characterized by the preference not to be and talk to people, not to act in case of troubles (items 7, 10, 11, 14). That could suggest some apathy or depressive signs in AD patients, despite the Zerssen Rating Scale not finding evidence of depression syndrome. However, we observed a significant link between Zerssen Rating Scale and SSPS-R total scores in both groups. Moreover, both scales share some items such as the preference to be alone, feelings of boring and uninteresting life. This suggests a possible association between alexithymia and affective disorders such as depression, but we cannot suggest that alexithymia can be completely subsumed within affective disorders. Alexithymia and depression are closely related constructs supported by inadequate emotional regulation strategies [65,66,67,68,69,70,71]. In addition, there is a wide literature about the presence of alexithymic traits in the individuals with suicidal behaviour [10,64] and apathy [14,15,44]. Nevertheless, the question still remains if alexithymia predisposes people to depression or at reverse or they co-exist, though most studies consider that alexithymia is not a product of depression [65,68,72,73]. In addition, sleep disorders such as sleep apnea, insomnia, or those in depression can affect cognitive performances, as well as they can be considered as a risk factor for dementia [74,75,76]. In our study, we did not observe depression in either subject group, but the control group remembered to dream more frequently than the AD group (the SSPS-R item 19). Unfortunately, we did not include an assessment of sleep quality or a history of sleep disorders for all participants and it can be considered as a limitation of the present study. It would also be interesting to observe the relationships between alexithymia and sleep disorders in future research on neurodegenerative diseases. Further studies will be necessary to put clearer the close constructs, considering the implications for treatment [77]. The alexithymic features in the patients with AD or other neurodegenerative disease should encourage more comprehensive therapeutic approaches, both pharmacological and no-pharmacological, that could prevent the development of more severe diseases and thus, improve the life quality of patients and their caregivers.

The present study does not confirm the previous results of other studies revealing great alexithymia assessed with TAS in AD patients [44,45,46], despite us being aware of SSPS-R test, used to assess alexithymia in our study, cannot completely be compared with other alexithymia scales. Yuruyen et al. [45] observed more alexithymic features assessed with TAS in mild AD and mild cognitive impairment (MCI) groups than in the subjective cognitive decline group, but they did not include a healthy control group. The research of Sturm and Levenson [44] studied only eight AD patients among other neurodegenerative diseases such as semantic or fronto-temporal dementias, observing that 80% showed high alexithymia levels. The research of Smirni et al. [46] also reported greater alexithymia total scores, but only higher scores on the TAS factor about the difficulty in identifying feelings in mild AD and MCI groups than those in the control group. In contrast, the remaining two factors of TAS (difficulty in describing feelings and externally oriented thinking) showed no significant group differences. They also found that TAS total score was negatively correlated with general cognition, attention, memory, and visual spatial constructive and executive abilities. In addition, the TAS factor about identifying feelings was also negatively correlated with general cognition, memory, and executive abilities in AD patients. Participants with AD and MCI showed a significant correlation between identifying feelings factor and long-term memory. They suggested that verbal memory troubles may hinder a patient’s ability to recall an association between a given sensation and the episodic experience of that sensation, thus leading to difficulty identifying feelings. Despite SSPS-R test not really being composed of items assessing the aspect of identifying feelings, our study cannot extend these results, because no correlations between alexithymia and cognition variables were found. In addition, we observed more difficulties in describing feelings in the AD patients than in the control group, as well as a greater externally oriented analytic mode of thinking in control group than in AD group.

In conclusion, the present results do not support the results of previous studies about alexithymia in AD. In our study, the patients with a mild AD show similar scores of alexithymia assessed with SSPS-R, but we observed that AD patients produced more alexithymic responses in some items of the SSPS-R test than the control group, particularly about difficulties to find the words to describe feelings, as well as difficulties of imagination capacity and externally oriented thinking. In addition, we only found an association between alexithymia and the mood state scores in both groups.

Finally, it is clear that clinically there is still much to be learnt about alexithymia and its relationship with related phenomena and symptoms of neurodegenerative diseases. Future studies with a longitudinal prospective design would be beneficial to test the stability of alexithymic responses in different stages of neuropathologies. We also believe that assessment of alexithymia may be a useful addition to the neuropsychological assessment of patients with brain damage because it could help to predict prognosis and response to specific therapies.

## Figures and Tables

**Figure 1 jcm-10-00044-f001:**
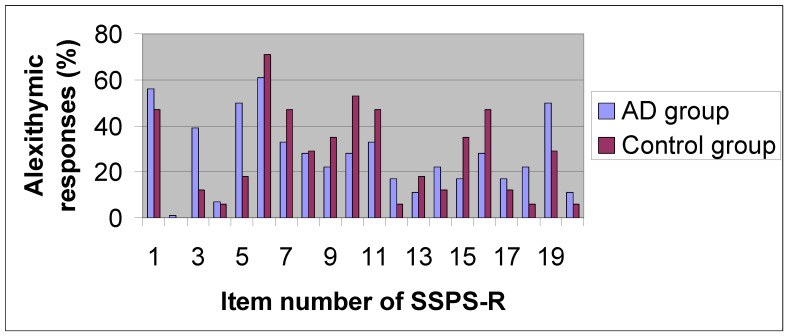
Alexithymic responses (%) of every Shalling Sifneos Psychosomatic Scale (SSPS-R) item in the AD and control groups.

**Table 1 jcm-10-00044-t001:** Means (M) and standard deviations (SD) of several demographic characteristics of the Alzheimer’s disease (AD) and control groups.

	AD Group	Control Group	
	M (SD)	M (SD)	*p* Value ^a^
Sample size (N)	27	27	
Gender (M:F)	8:19	13:14	0.062 NS
Age (years)	79.17 (7.43)	76.59 (9.43)	0.16 NS
Education years	8.94 (1.64)	9.26 (1.11)	0.18 NS
Clinical Dementia Rating System (CDR)	1.4 (0.23)		
Years of illness	4.1 (1.22)		

^a^*p* values referred to comparison between control and AD groups. NS: not significant (*p* > 0.05).

**Table 2 jcm-10-00044-t002:** Means (M) and standard deviations (SD) of neuropsychological tests in the AD and control groups.

	AD Group	Control Group	
	M (SD)	M (SD)	*p* Value ^a^
**Mini-Mental State Evaluation (MMSE)**	20.28 (4.61)	29.18 (1.13)	0.0001 *
**Free/cued recall test (Grober and Buschke verbal memory test):**			
**-short delay free recall**	4.94 (4.82)	32.28 (4.02)	0.0001 *
**-short delay free and cued recall**	16.33 (11.81)	43.67 (2.15)	0.0015 *
**-total recognition**	6.47 (1.27)	13.48 (0.83)	0.0012 *
**-long delay free recall**	1.33 (1.81)	10.01 (1.24)	0.0001 *
**-long delay free and cued recall**	3.67 (3.96)	14.92 (0.21)	0.0001 *
**Verbal naming test (DO80)**	71.29 (5.95)	79.46 (1.20)	0.032 *
**Zerssen Rating Scale**	11.63 (7.39)	12.18 (6.12)	0.066 NS
**Alexithymia test: the Shalling Sifneos Psychosomatic Scale (SSPS-R)**	5.44 (2.75)	5.35 (2.76)	0.93 NS

^a^*p* values referred to control and experimental groups. ***** Significant difference (*p* < 0.05). NS: not significant (*p* > 0.05).

**Table 3 jcm-10-00044-t003:** Correlations of Shalling Sifneos Psychosomatic Scale (SSPS-R) scores with cognitive and mood variables in the AD and control groups.

	AD Group	Control Group
**Age**	0.128 (*p* = 0.61)	−0.249 (*p* = 0.22)
**Education Years**	0.013 (*p* = 0.55)	0.040 (*p* = 0.74)
**Mini-Mental State Evaluation (MMSE)**	−0.155 (*p* = 0.54)	0.232 (*p* = 0.37)
**Free/Cued Recall Test (** **Grober and Buschke verbal memory test):**		
**-short delay free recall**	0.015 (*p* = 0.95)	0.132 (*p* = 0.99)
**-short delay free and cued recall**	0.091 (*p* = 0.72)	0.116 (*p* = 0.84)
**-total recognition**	0.098 (*p* = 0.85)	0.093 (*p* = 0.94)
**-long delay free recall**	−0.115 (*p* = 0.65)	−0.153 (*p* = 0.54)
**-long delay free and cued recall**	0.085 (*p* = 0.74)	0.110 (*p* = 0.87)
**Verbal Naming Test (DO80)**	−0.645 (*p* = 0.07)	−0.214 (*p* = 0.33)
**Zerssen Rating Scale**	0.644 (*p* = 0.01) *****	0.692 (*p* = 0.01) *****

***** Significant difference (*p* < 0.05).

**Table 4 jcm-10-00044-t004:** Comparison of alexithymic responses (%) in each Shalling Sifneos Psychosomatic Scale (SSPS-R) item between the AD and control groups.

Item Number	SSPS-R Items	Alexithymic Responses	Alexithymic Responses (%) of Ad Group	Alexithymic Responses (%) of Control Group	*p* Value ^a^
1	It is easy to describe symptoms or complaints rather than feelings	Yes	56	47	0.561
2	It is important to find out how one feels about people	No	1	0	0.981
3	It is hard to use words to describe feelings	Yes	39	12	0.0163 *
4	I think that feelings are what make life worthwhile	No	7	6	0.966
5	I lack imagination	Yes	50	18	0.0124 *
6	I spend much time daydreaming	No	61	71	0.582
7	When I am mad I don’t think, I take action	Yes	33	47	0.0221 *
8	I like movies with action rather than psychological dramas	Yes	28	29	0.893
9	When in conflict I prefer to act quickly, rather than to think about it	Yes	22	35	0.487
10	When in trouble I don’t like to act	No	28	53	0.0182 *
11	I like to be precise and to describe everything in detail	Yes	33	47	0.0211 *
12	I don’t care to describe details but rather I prefer to examine how I feel	No	17	6	0.715
13	I have difficulty to communicate with people	Yes	11	18	0.527
14	I prefer to be alone rather than interact with people	Yes	22	12	0.0236
15	I always pay attention to my surroundings rather than concentrate on how I feel	Yes	17	35	0.0204 *
16	When I hear suspicious noises at night I don’t ever concentrate on how I feel	Yes	28	47	0.715
17	I find life boring most of the time	Yes	17	12	0.587
18	I cannot visualize circumstances which upset me	Yes	22	6	0.0231 *
19	I dream rarely	Yes	50	29	0.0216 *
20	I like people better than things	No	11	6	0.794

For alexithymia individuals assessed using the Shalling Sifneos Psychosomatic Scale (SSPS-R): questions 2, 4, 6. 10, 12. 20 are false ones and should be answered No. All other questions should be answered Yes. Thus, a score of 10–20 should point to alexithymic characteristics. ^a^
*p* values referred to control and experimental groups. * Significant difference (*p* < 0.05). NS: not significant (*p* > 0.05).

## Data Availability

All data of study are available in Department of Neurology and Unit of Neuropsychology at CHU of Poitiers, France.

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
