# Peer review of "Alexithymia in Alzheimer’s Disease"

_jcm, 2020, doi:10.3390/jcm10010044_

Round 1

Reviewer 1 Report

Few investigations have studied the alexithymia in AD. Thus this research gives new data regarding the problem. In my opinion this paper is acceptible to be published in research journal. And I can make just minor suggestions about improvement.

First of all English editing is needed. Besides, there are some mistakes of typing, missed letters.

Every table should contain explanation of all abbreviations.

In Fig. 1 it would be more correct to use columns rather than a curve as the data on different items of SSPS-R are presented.

Author Response

Dear reviewers and editor:

Thanks a lot for your help, it was very useful.

The authors have trying to answer all comments raised by the reviewers, which they are very useful to improve the manuscript. We are very grateful.

All author details on the revised version are correct (in red color) and all authors are agreed to authorship and the order of authorship for the manuscript.

Reviewer 1:

- The language expert (Ms. Lucy Wanja Maina from Teyde company: [email protected]) has corrected the English language in manuscript.

- We've included the explanation of all abbreviation in every table.

- We've used columns in Fig. 1

Reviewer 2 Report

The work is well done. I suggest a more detailed description of background because it is not clear the concept of alexithymia. 

Author Response

Dear reviewers and editor:

Thanks a lot for your help, it was very useful.

The authors have trying to answer all comments raised by the reviewers, which they are very useful to improve the manuscript. We are very grateful.

All author details on the revised version are correct (in red color) and all authors are agreed to authorship and the order of authorship for the manuscript.

Reviewer 2:

- We've added more details about the concept of alexithymia. 

Reviewer 3 Report

Alexithymia is conceptualized as a cluster of cognitive traits, which include difficulty 31 identifying feelings, describing feelings to others, externally oriented thinking, and limited imaginative capacity. Several studies have linked alexithymia to cognitive functioning, observing  greater alexithymia scores associated with poorer cognitive abilities.

The present results do not support the results of previous studies about  alexithymia in AD. In our study, the patients with a mild AD show similar scores of alexithymia but we observed that AD patients produced more alexithimic responses in some items of SSPS-R test than the control group, particularly about difficulties to find the words to describe feelings, as well as difficulties of imagination capacity and externally oriented thinking.

I think that the paper in vert interesting and well written article.

In the method section, I would consider a through history of sleep disorders for both patients and controls.

From the literature It is known that sleep disorders are involved in  the hippocampal-frontal lobe connection mechanisms. Not taking them into account may mean blurring the differences between the two groups.

Add these references:

Buratti L, Luzzi S, Petrelli C, Baldinelli S, Viticchi G, Provinciali L, Altamura C, Vernieri F, Silvestrini M.Obstructive sleep apnea syndrome: an emerging risk factor for dementia. CNS & Neurological Disorders - Drug Targets 2016;15:678-82.

Buratti L, Viticchi G, Baldinelli S, Falsetti L, Luzzi S, Pulcini A, Petrelli C, Provinciali L, Silvestrini M. Sleep Apnea, Cognitive Profile, and VascularChanges: An IntriguingRelationship. Journal of Alzheimer’sDisease2017;60:1195-1203.

Author Response

Dear reviewers and editor:

Thanks a lot for your help, it was very useful.

The authors have trying to answer all comments raised by the reviewers, which they are very useful to improve the manuscript. We are very grateful.

All author details on the revised version are correct (in red color) and all authors are agreed to authorship and the order of authorship for the manuscript.

Reviewer 3: 

- You are completely right. Sleep disorders can affect cognitive performances and can be considered as a risk factor for dementia. Unfortunately, we did not include an assessment of sleep quality or a history of sleep disorders for all participants in our study and it is a limitation of the present study.

We've included this limitation in discussion and we've added these references (Buratti et al., 2016; 2017).

Many thanks for the review.